# Lived Experiences of Hospitalized COVID-19 Patients: A Qualitative Study

**DOI:** 10.3390/ijerph182010958

**Published:** 2021-10-18

**Authors:** Montserrat Venturas, Judith Prats, Elena Querol, Adelaida Zabalegui, Núria Fabrellas, Paula Rivera, Claudia Casafont, Cecilia Cuzco, Cindy E. Frías, Maria Carmen Olivé, Silvia Pérez-Ortega

**Affiliations:** 1Hospital Clinic Barcelona, 08036 Barcelona, Spain; mventura@clinic.cat (M.V.); jpratsc@clinic.cat (J.P.); equerol1@clinic.cat (E.Q.); azabaleg@clinic.cat (A.Z.); privera@clinic.cat (P.R.); casafont@clinic.cat (C.C.); ccuzco@clinic.cat (C.C.); cfrias@clinic.cat (C.E.F.); 2School of Nursing, University of Barcelona, 08036 Barcelona, Spain; nfabrellas@ub.edu (N.F.); olivecarmina@gmail.com (M.C.O.)

**Keywords:** care needs, patient experience, phenomenology, coronavirus, qualitative study

## Abstract

The COVID-19 pandemic has resulted in many hospitalized patients and deaths worldwide. Coronavirus patients were isolated from their relatives and visits were banned to prevent contagion. This has brought about a significant change in deeply rooted care habits in Mediterranean and Latin American countries where the family normally accompanies vulnerable hospitalized patients. The aim of this qualitative study was to examine the hospitalization experience of COVID-19 patients and their family members. A phenomenological qualitative approach was used. Data collection included inductive, in-depth interviews with 11 COVID-19 hospitalized patients. The mean age of patients was 55.4 years and 45% were female. Nearly 50% required Intensive Care Unit (ICU) admission. Ten meaningful statements were identified and grouped in three themes: Positive and negative aspects of the care provided, the patient’s perspective, and perception of the experience of the disease. In conclusion, COVID-19 patients, aware of the severity of the pandemic, were very adaptable to the situation and had full confidence in health professionals. Patient isolation was perceived as necessary. Technology has helped to maintain communication between patients and relatives.

## 1. Introduction

At the beginning of 2015, the Global Challenge Foundation, in cooperation with the University of Oxford, published a first report that highlighted natural risk factors, which, if they occur, might cause such significant damage worldwide and that they could even endanger human survival. The report principally examined 12 risks, including the risk of pandemics, and warned of the enormous impact that these may have on the modern world and the catastrophic consequences they could imply [1].

Throughout history, some pathogens have put the viability of the human species at risk: the Black Death in the Middle Ages, for example, wiped out more than half of the European population. The so-called “Spanish” flu of 1918 resulted in around 500 million infections and 50 million deaths. Since the beginning of the 21st century, there have been several pandemics, such as SARS, Ebola, and MERS, and now the current pandemic caused by the ongoing SARS-CoV-2 coronavirus (COVID-19) [1,2].

The first cases of people affected by SARS-CoV-2 were detected in China at the end of 2019, with the city of Wuhan as the epicenter. Since then, this new virus has caused more than 171 million infections and more than 3.5 million deaths worldwide as of May 2021 [3].

Due to the high rate of infection and the virulence, about 20% of infected people require hospital admission for respiratory failure [4]. This has led to an increase in the demand for primary care and hospitalization, triggering, in some countries, the saturation of health systems, which have been forced to double or even triple the supply of beds, especially in intensive care [5].

In Spain, due to the saturation of the health system, the number of inpatient beds and intensive care units (ICUs) was expanded, converting offices, day centers, and even hospital corridors; at the same time, hotels and sports facilities close to large hospitals are being set up to cope with the substantial increase in infected patients. The Hospital Clínic of Barcelona, for example, received more than 2400 inpatients, including 267 ICU patients, from mid-March to mid-May 2020. In addition, a health hotel near the hospital was opened to attend more than 500 patients with moderate symptoms or in the recovery phase before discharge. During this time, 80% of hospital rooms were destined for COVID-19 patients [6].

This unprecedented healthcare crisis has forced governments of the affected countries to take extreme public health measures, such as quarantine and social distancing, to prevent the spread of the virus according to the recommendations of the World Health Organization (WHO) [7].

These measures clash head-on with the concept that, especially in cultures, such as the Mediterranean and Latin America, care must be offered to loved ones who are in a situation of vulnerability and/or disease.

Care involves patterns of understanding and gender behavior, culture, and society, and therefore, nursing practices involve knowledge and relationships with diverse cultures and societies [8,9]. In Spain, progress has been made in recent years towards patient- and family-centered care, such as allowing patients to be accompanied by their close relatives during hospitalization, including in intensive care units and, especially, in end-of-life processes [10].

Mishel’s theory of uncertainty in the face of disease explains how people experience uncertainty in a situation of illness [11]. Interventions are focused on achieving a harmonious adaptation and a new perspective on life. In this way, it is possible to face the disease and integrate uncertainty as inherent in the health situation and experience uncertainty as a positive force that offers multiple opportunities to achieve a good mood in patients.

The introduction of social distancing, total confinement, and quarantine measures causes disruption of the culture of care in the family, which is even more evident during periods of hospitalization, as patients have been isolated from their families and no visitors are allowed. Communication with family members was through patients’ cell phones when possible. Medical information for ICU patients was provided by doctors and nurses. However, patients on the general ward were usually kept informed by medical students. At the end of March, to mitigate anxiety and uncertainty in patients and relatives, a program was started called “shortening distances”, where psychologists provide technology and support for patients to communicate with family members.

In previous epidemics, such as SARS, anxiety and depressive symptoms were described in approximately 35% of survivors [12]. During the same period, other authors studied the different forms of post-traumatic stress that patients developed, depending on their severity, obtaining results similar to those studied in other diseases [13].

A recent review by Brooks et al. shows how the quarantine used in some 21st century epidemics (SARS-CoV, MERS-CoV, influenza A/H1N1, and Ebola) affected humans psychologically, causing a high prevalence of mental symptoms [14].

Similarly, the current SARS-CoV-2 pandemic has triggered a wide variety of psychological problems, such as panic disorder, anxiety, and depression [15,16]. However, no qualitative studies have been found that describe patients’ experiences after an epidemic or pandemic. These experiences could differ depending on the disease severity and patient experience of hospitalization for COVID-19. It is therefore necessary to study the experience of this care crisis to acquire knowledge for future situations with similar epidemiological and clinical characteristics. In addition, these studies will provide reliable and necessary information to adapt the elements of current care models, with the aim of ensuring early, organized, safe, and evidence-based action for future situations.

The aim of this study was to examine and describe the experience of patients with COVID-19 in a high technology hospital.

## 2. Materials and Methods

### 2.1. Design

We used a qualitative, phenomenological methodology [17,18], seeking to understand the lived experience of COVID-19 patients admitted to a high technology hospital.

We followed the Consolidated Criteria for Reporting Qualitative Studies (COREQ) 32-item checklist [19].

### 2.2. Selection of Participants

Patients gave oral consent, as authorized by the Hospital Clinic Research Ethics Committee due to the pandemic. Consent was obtained at the beginning of the interview and was witnessed by two researchers and an external witness. All informed consents were recorded and stored as confidential documents together with the document signed by one of the researchers and the witness.

A purposive sample of patients was recruited (*N* = 11) until data saturation was reached. We included patients with different hospitalization profiles: intensive care unit, the general ward, and health hotels, to minimize the risk of bias. Recruitment was made with the aid of nurses working on COVID-19 units.

### 2.3. Data Collection

To generate information, semi-structured interviews were conducted by two researchers (JP and SP), in accordance with the study objectives. They were carried out until data saturation [20].

Data were collected from April to June 2020, coinciding with the first wave. Due to the pandemic and containment situation, interviews were conducted via telephone, with loudspeakers, to perceive different expressions, tones of voice, or silences. Interviews were carried out when patients were at home after overcoming the critical period of the disease.

An analysis of the discourse was made from the literal transcription of the interview and the observations obtained by both researchers to show the why and what for of what was said and thereby understand the patient’s overall experience of the disease [17].

In all phases of the research, triangulation of subjects and researchers was made to increase the validity of the results and thus obtain a greater understanding of the lived and studied reality [21].

The questions were developed by the research team. We asked participants about their experience of isolation during admission, positive and negative experiences of the whole process, and the information received. Finally, we asked an open question so that participants could express whatever they wanted.

### 2.4. Data Analysis

Interviews were digitally recorded, transcribed verbatim, and analyzed using the Linsdeth and Norberg (2004) three-step phenomenological-hermeneutic method. This analytical technique involves ‘naive reading’ of transcripts to establish a sense of the data (reading and re-reading transcripts and listening to interviews with an ‘open mind’), structural analysis (initial identification and formulation of themes and sub-themes that convey a sense of the lived experience, in everyday language), and comprehensive understanding. This final phase involves a critical, in-depth interpretation of the data as a whole and reflection, revision, and reconsideration of the themes identified during the analytical process [22].

All analyses were performed individually by two researchers (MV and CC), who shared and discussed their experiences and opinions regarding data collection and analysis.

The rigor of the study was ensured through the following evaluation criteria: creditability, transferability, dependability, and conformability [23].

### 2.5. Ethical Considerations

The Hospital Clinic of Barcelona Research Ethics Committee at (HCB/2020/0518) approved the research project. The study was conducted in accordance with the Declaration of Helsinki [24]. We obtained oral informed consent prior to all interviews. To protect participants’ confidentiality, no records, including interview transcripts, noted their names or other identifiers. We presented all results anonymously and limited data access exclusively to four researchers.

## 3. Results

### 3.1. Participant Characteristics

Eleven patients participated in this study. Their characteristics are shown in Table A1.

The mean age was 55.4 years and 45% were female. Nearly 50% required ICU admission.

Participants were selected to reflect different profiles with respect to hospitalization: ICU, general ward, and health hotel. Almost half the patients were admitted to the ICU. The patients sample is shown in Table A2.

The meanings derived from the meaningful statements and rewordings were classified into thematic groups. Nine meaningful statements were identified and grouped in three themes: positive and negative aspects of the care provided, patients’ hospitalization experience, and perception of the experience of the disease.(Table A3)

Positive and negative aspects of the care provided included uncertainty, information and communication, isolation, confidence, and flexibilization. Patients’ hospitalization experience included the patient’s adaptation to the situation and human and material resources. The perception of the experience of the disease included the hospital process and abandonment.

### 3.2. Theme 1. Positive and Negative Aspects of the Care Provided

#### 3.2.1. Isolation

In this situation, family visits were not allowed, and health professionals minimized visitors entering patient rooms. However, for most patients, it was not a negative aspect.

“I had never been hospitalized, the first time in 63 years, when I had already been confined at home for days and I’m still... I’m not a person who needs a lot of people around me. I know that my family was missing me and so did they, but it wasn’t difficult for me”.(2 Participant)

Mobile technology also helped patients maintain contact with family members and not feel alone.

“even if you are isolated, as you’ve got your mobile and the nurses come in from time to time to check your temperature, you’re not really alone... I guess that having my mobile and not being a severe case, I could get up and I didn’t need oxygen, which also made things easier”.(5 Participant)

#### 3.2.2. Confidence

Most patients reported that, although before hospital admission they had a lot of uncertainty due to the disease, once admitted, they were confident and felt safe.

“I see that the level of professional competence makes you feel safe being in the hospital”.(4 Participant)

“I’ve never lived the experience from inside a hospital, first you may think it’s chaotic, but then you see that everything is coordinated. All the professionals know what has to be done and everyone is coordinated, so you feel confident”.(2 Participant)

#### 3.2.3. Uncertainty

During hospitalization, trust in health centers and their professionals meant that the uncertainty occurred mainly due to the disease evolution.

“Each day getting worse and worse, and of course I said:—my God, it seems I’m not getting better... [cries] I just asked for my life to be saved, because I have a three-year-old girl”.(9 Participant)

#### 3.2.4. Information and Communication

Due to isolation and because family members could not be with patients, communication between professionals and family members was diminished. The lack of doctors meant that medical students, who had not seen the patient, had to telephone families to give information due to this situation, and the patients stated that there was little information given to the family.

“little information, my children told me they had little information. Of course, they could not visit either, but they received little information”.(6 Participant)

However, conscious patients received medical information personally and found it satisfactory.

“the information was correct, at least what they gave me... I never had the feeling of not knowing how everything was, or what was happening. As for my relatives, they got more information from me than from the hospital. My wife received a call only once to tell her that I had just come out of the ICU, just once”.(4 Participant)

In all the interviews, the contribution of new technologies as a facilitator in the communication between patients and families was mentioned.

“Except for the days I was in the ICU sedated, you communicate with the mobile, the mobile has helped a lot”.(2 Participant)

#### 3.2.5. Flexibility

The patients said they saw nurses’ difficulties in adapting to the new situation.

“I saw all the difficulties that nurses had in caring for patients. The fact that we were in rooms with a maximum of two people made the tasks very difficult. I imagine that larger rooms, with more patients together would have been better... hospitals, at least this one, have demonstrated an ability, a flexibility, an extraordinary ability to adapt to the new situation that was really impressive”.(4 Participant)

### 3.3. Theme 2. Patients Hospitalization Experience

#### 3.3.1. Patient Adaptation to the New Situation

All patients reported being grateful for the care received; however, it was gratitude for the effort of the professionals as patients were aware that care was conditioned by the pandemic. They stated that they had adapted to the situation, they knew that nurses could not always come immediately, and they did not demand it.

“I wish there had been more people, because as I say, the first days, above all... uh, things were done late, but I imagine that’s why. The procedures didn’t let them come before”.(11 Participant)

“all very nice, always very dedicated, and although I understood that the security measures meant that nurses did not come in to the rooms all the time, but from time to time they opened the door, looked in and asked how you were, always attentive to what you wanted. I would say very dedicated”.(5 Participant)

Patients also expressed concern that they could not see the face of nurses caring for them but resigned themselves and appreciated the treatment received.

“the professionals gave us a lot of love, when they were taking our blood pressure. All “disguised”, you couldn’t see their faces, you only heard the voice. They were very sweet and made us feel secure and tranquil. You got the feeling that you weren’t alone. They’re there”.(3 Participant)

“for me this is very important, to know the people, in the way they have treated you, and to be able to thank them, you already do it but they have their face [laughs] covered and you hardly see them, just the eyes a little”.(10 Participant)

#### 3.3.2. Human and Material Resources

Several patients expressed solidarity with health workers due to the lack of equipment and professionals during the first wave of the pandemic.

“I only saw that they were drowning in the situation and that there was a lack of care, other nurses didn’t have time. Since it was undressing, dressing, getting into bed, this takes a while, more nurses were needed”.(6 Participant)

“the nurses should have had more material, they had little and each day was different, this didn’t make us feel secure when they were working with fear”.(3 Participant)

### 3.4. Theme 3. Perception of the Experience of Disease

#### 3.4.1. Hospital Process

The different perceptions that the interviewees have of the ICU are related to the degree of knowledge of the ICU and the severity of their illness at ICU admission. Several interviews showed that patients suffered a type of amnesia about their ICU stay.

“I didn’t know I was going into the ICU, I was in intermediate care. I had walked into the hospital and was sent to intermediate care. I guess I ran out of air, and they took me to the ICU: I woke up in the ICU, everything was shining, it was very modern, you were very protected, it looked like 2001 A Space Odyssey. I didn’t know if I was in heaven or hell... someone came and started asking me questions... who I was, what my name was, I didn’t understand the reason for the questions. I missed this part of the disease”.(2 Participant)

“I remember walking into the hospital... I had my chest X-ray, and I don’t remember anything else. And I know I was awake until I was intubated, and I don’t remember that I had talked to the doctor, who asked me for consent to be intubated... I don’t remember anyone. I’ve erased that from my mind. There are things that your mind erases and it’s over. [Silence]”.(1 Participant)

Patients remembered waking up in the ICU as a strange situation after their recovery.

“The days I was awake in the ICU were very oppressive, it felt as if I was in a movie in the third dimension, I didn’t believe what was going on... had very weird feelings, I don’t know, sounds [silence] I guess were all from my head”.(1 Participant)

#### 3.4.2. Abandonment

After waking up in the ICU, some patients had delirium, such as this patient who believed she had been abandoned by her family.

“At first I took it hard. It’s because I didn’t know I had this. And I remember nothing from the last few days I had it. So what happened, I woke up and found myself alone, without my children, who didn’t know what I had. I thought they had left me (choking, emotional voice). I had a hard time in the sense that I thought I had been left there alone”.(6 Participant)

Patients expressed dual feelings: they felt abandoned by their family, but at the same time they expressed feelings of security by avoiding infecting their relatives.

“I couldn’t see my children, I thought they had abandoned me. Then I understood it, although I was grieving inside, but I also thought what if they catch it, so I have to put up with it, in the end I understood”.(6 Participant)

## 4. Discussion

Three main themes—positive and negative aspects of care, patients’ hospitalization experience, and perception of the experience of the disease—were described by patients admitted to the ICU during the COVID-19 pandemic.

The security of the Spanish health system was overwhelmed by the rapid increase in patients requiring ICU admission. This meant many patients were admitted when symptoms were already severe. The patients interviewed had different profiles depending on the unit of admission and although in the interviews they explained their experience throughout the hospital process, those who were in the ICU manifested other aspects as a result of amnesia and delirium.

Isolation was not experienced as negative by patients but as something inevitable. However, health professionals expressed suffering due to the isolation and loneliness of patients [25]. In our study, feelings of loneliness may not have been revealed as patients experienced isolation as necessary. The use of mobile phones together with trust in professionals contributed to isolation not being perceived as something negative.

Mobile devices and new technologies facilitated the communication of patients with their relatives, and in turn, allowed professionals to inform the family about their clinical status. Akgün explains how, in this pandemic, two-way communication may help alleviate the fear of abandonment and the feelings of uncertainty that many patients and families experience during COVID-19 hospitalizations [26,27]. In previous crises, such as SARS, information was already communicated to family members with the use of new technologies that helped combat anxiety [12]. Health professionals, accustomed to a more direct and personalized contact, have had to adapt to this new form of communication [28,29].

Reports show the uncertainty that this pandemic has generated in healthcare professionals, in contrast to the feelings of patients in their interviews [30,31,32]. Although it is true that some patients refer to uncertainty regarding the evolution of the disease, most expressed peace of mind and confidence when they were hospitalized and controlled in the ICU. This may also be influenced by the fact that the most severe patients did not remember their admission to the ICU and, therefore, may not have experienced such uncertainty.

One of the features of the interviews was confidence and adaptation to the situation, despite patients stating that they were not always cared for as they thought they should be due to the shortage of human and material resources caused by the pandemic. The patients were very positive about the care received and the generosity of the professionals. Spain has one of the lowest ratios of nurses worldwide, with 5.9 nurses per 1000 inhabitants and one nurse for every two critical patients; although this ratio is variable, this lack of human resources became more latent during the pandemic [33]. All patients adapted rapidly to the new circumstances, with no differences observed according to age or sex. The study reflected the ability of humans to adapt and face up to adverse events by the use of many physical and psychological resources in order to overcome the situation [34].

In the Hospital Clinic of Barcelona, the number of ICU beds was rapidly increased, professionals were relocated (outpatient services, elective activity, diagnostic tests...), and specific jobs were created under the aegis of the state of alarm (health auxiliaries) and holidays and free days were temporarily cancelled.

Gratitude is inherent in vulnerable human beings and all participants expressed expressions of gratitude to the professionals and health system and for the medical treatment received [35]. Something similar was reflected in the study by Niner et al. on the birth experiences in Australia of Karen and Kayin women displaced from Burma [36]. The women they interviewed expressed gratitude for a variety of circumstances (safe haven, safe environment, care provided, and postpartum support) even though many had experienced suboptimal care and lack of autonomy. Luhmann explains why humans need to trust to be safe. This dose of confidence reduces the complexity of the action and helps people to think that everything will be fine [37]. The gratitude shown by the participants was interpreted as trust in the professionals and allowed them to feel safe and maintain the peace of mind necessary to balance their mental health.

Patients admitted to the ICU experienced amnesia upon admission and during intubation and, subsequently, delirium that persisted for days after extubation. This situation, common to other diseases in some patients, occurred more frequently in patients with severe COVID-19. Multiple neurological manifestations have been described, mainly in patients with very severe COVID-19 [38,39]. One participant felt her family had abandoned her, a feeling that lasted from waking up in the ICU to being discharged home.

Public health emergencies, such as this pandemic, are stressful times for individuals and communities, creating social stigma, as reflected by some participants [7]. Sahoo, in a narrative of surviving COVID-19 patients, found that, due to isolation, the fear of death, and associated stigma, many patients with COVID-19 infection suffered mental anguish [40]. The fear marked by the disease and its contagion may be seen as a stigma of the disease.

This study has reflected a reality not expected by researchers: patients did not feel the hospital experience was negative. Maunder et al. studied the experience of patients during the SARS outbreak in Toronto, and found fear, loneliness, boredom, and anger [41]. However, the participants expressed confidence, security, adaptation, and little uncertainty, and did not manifest fear or loneliness. This might be because everyone successfully overcame the disease, and the interviews were conducted when they were already at home. In both studies, interviews were conducted approximately one month after overcoming the disease. It will be interesting to assess the consequences on the mental health of people admitted to the ICU due to severe COVID-19, in search of psychiatric symptoms or post-traumatic stress disorder, as already seen after the SARS outbreak [13,42].

### 4.1. Limitations

The study had some limitations. Firstly, it was carried out in a single center although, due to the pandemic, the treatment given to hospitalized COVID-19 patients was similar to that of other centers: isolated and without family visits. Therefore, the results presented can guide and be useful in the preparation of future studies.

Second, our sampling strategy was based on conducting interviews until data saturation. To the surprise of the researchers themselves, this saturation was reached very quickly, finally leaving a small study sample. The authors attribute this mainly to three reasons. Firstly, the general feeling of gratitude and recognition for the efforts made by the professionals in charge of their care. Secondly, the fact that the method of data collection chosen was through semi-structured interviews may have limited in some way the expression of thoughts, feelings, or ideas that are on a more unconscious plane or that require deeper elaboration. Thirdly, during the preparation of the study design, the researchers expected to find significant differences between patients depending on the units to which they were admitted, something that was not extracted later during the conduct of the interviews since the patients talked about their hospital experience as a whole, and not segmented by the different phases of the process.

Thirdly, it was decided to carry out the interviews by telephone, as there was a state of alarm, and this was the best way to guarantee the safety (by avoiding travel) of all involved. This may represent a bias as we were not able to observe the interviewees (thus losing information related to non-verbal language and the possibility that people will be more spontaneous), but this interview method was chosen in an attempt to capture the lived experience and the emotions felt in the most reliable and vivid way possible, while trying to avoid false memories or other possible errors derived from the passage of time. The objective of the study was to elucidate the needs at the time of hospitalization to provide the best possible care during this experience to our future patients. The researchers do not rule out conducting a more in-depth study that includes interviews at one or two years to check the evolution of the subjects and the long-term impact of the experience.

Despite the limitations, we believe our results may be extrapolated, since the ignorance of the disease, the fear of infection and infecting other, the measures established to combat the pandemic, and the health and socioeconomic consequences have been very similar around the world.

### 4.2. Implications for the Future

In-depth determination of the perceptions and experiences of patients about COVID-19 hospitalization and the isolation it involves allows nursing actions aimed at minimizing negative feelings and promoting a more satisfactory evolution of their illness to be established, contributing to the humanization of care.

The feelings of patients should serve to find points of improvement in health care in future pandemic waves.

We highlight the usefulness of technologies, such as video conferencing, to improve communication between patients and families, whose use could be incorporated and enhanced in usual practice. Studies of the psychological status of patients and possible manifestations of post-traumatic stress related to COVID-19 hospitalization during the pandemic crisis are required.

## 5. Conclusions

We identified three main themes: positive and negative aspects of the care provided, patients’ hospitalization experience, and the perception of the experience of the disease in patients admitted to the ICU during the COVID-19 pandemic.

This study has reflected a reality not expected by the researchers: patients did not feel the hospital experience was negative. COVID-19 patients, once admitted, and aware of the severity of the pandemic, adapted to the situation and showed confidence in health professionals and the treatments offered.

The isolation of patients was perceived as necessary. New technologies helped to maintain communication between patients and families.

The COVID-19 pandemic has highlighted the importance of a good public health system. Solutions must be sought to already chronic problems in the health system, such as the lack of nurses, low salaries, and temporary recruitment, which have become even more evident in this crisis. Specialized nurses and continuity in care permit trust to be increased and a greater bond between the interdisciplinary team and the patient created.

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
