# Peer review of "Lived Experiences of Hospitalized COVID-19 Patients: A Qualitative Study"

_ijerph, 2021, doi:10.3390/ijerph182010958_

Round 1
Reviewer 1 Report
Dear Authors,
Thank you very much for this interesting work. I think it is a well written manuscript. I have however some suggestions for its improvement:
- in the Introduction, lines 26-31 - reference should be given;
- lines 50-54 - your are stating "our hospital ...." - it would be better to write directly the name and place of the hospital;
- line 107-108: "... patients signed the oral consent..." - it does not make sense. If signed, it is written consent...;
- In results, Theme 2 - Could you think about the different name for it? Actually, all of the results it is a 'patient perspective'... therefore, it would be good to make it more clear;
- discussion, lines: 324-326 - here again you are writing "... our hospital...", please, write in more formal style;
- in the conclusions, could you create more practical implications for the future?
Author Response
We thank you for reviewing our article.
- We have added the bibliography corresponding to line 27, and to the data on lines 33-35.
- The name of the Hospital has been identified
- We have explained extensively how the consent was given, which was oral, recorded and witnessed, as authorized by the ethics committee.
- The name of the second topic has been modified for clarity.
- We have expanded the discussion of the most salient points, and have incorporated the limitations of the study, sampling strategy, and future implications.
Reviewer 2 Report
The goal of the study is sound and useful to potential readers. The study is however heavily biased by the very limited number ot the patients interviewed, moreover in different environments (ICU, hospital wards, COVID hotels) and by the fact that a single institution has been taken into account.
Consequently, the data reported are merely occasional an may not be generalized to a wider community of patients and health care institutions, and this represents a major limit of the study.
Author Response
We thank you for reviewing our article.
The study sample was purposively collected, as participants are recruited in most qualitative studies. Participants who were expected to contribute satisfactory results to the study were purposively selected. The sample size was determined by data saturation. We have explained this point at length in the article as suggested to us. The aim of our research is in-depth qualitative inquiry, not generalization of data.
However, we widely believe that different subjects worldwide may have a similar experience.
Limitations of the study and future implications have been included.
We add some literature reviewed in relation to sample size
Patton MQ. Qualitative research and evaluation methods. 3rd Sage Publications; Thousand Oaks, CA: 2002.
Sandelowski M. Sample size in qualitative research. Res Nurs Health. 1995;18:179-83
Reviewer 3 Report
The study is well-prepared in theoretical, methodological and research terms. Qualitative studies of the care-related needs and experiences of patients hospitalised during the COVID-19 pandemic in the phenomenological context provide a suitable perspective for investigations into new phenomena that had not been present before. The selection of individuals to the examined group is justified. The research results are highly interesting due to the new circumstances of providing hospital care during the COVID-19 pandemic and the need to maintain social distance by the informal caregivers of hospitalised patients. The study results relating to three important subjects and their subtopics: Positive and negative aspects of the care provided (Isolation, Confidence, Uncertainty, Information and communication Flexibility), Patient perspective (Patients’ adaptation to the new situation, Human and material resources), Perception of the experience of disease (Hospital process, Abandonment), provide important clues for healthcare organisers and an insight into the expectations of hospitalised patients and their families in the process of therapy and care. The discussion of study results is well-argued. However, study limitations are not indicated. The conclusions are well-defined, although they lack a description of the practical implications of the results. The greatest value of this research project lies in its innovative nature of showing the viewpoint of hospitalised patients and their families under the threat of the SARS-CoV-2 epidemic.Author Response
We thank you for reviewing our article.
We have expanded the discussion on those points that are most salient, and have incorporated the limitations of the study and future implications.
Round 2
Reviewer 2 Report
minor revisions in the text